# Prevalence and predictors of *Aspergillus* seropositivity and chronic pulmonary aspergillosis in an urban tertiary hospital in Sierra Leone: A cross-sectional study

Sulaiman Lakoh[1,2,3]*, Joseph B. Kamara[1,2], Emma Orefuwa[4], Daniel Sesay[2], Darlinda F. Jiba[2], Olukemi Adekanmbi[5,6], Gibrilla F. Deen[1,2], James B. W. Russell[1,2], Abubakarr Bailor Bah[1,2], Maxwell Joseph Kargbo[1], Emmanuel Firima[7,8,9,10], George A. Yendewa[11,12,13], David W. Denning[4,14]

1 College of Medicine and Allied Health Sciences, University of Sierra Leone, Freetown, Sierra Leone, 2 Ministry of Health and Sanitation, Government of Sierra Leone, Freetown, Sierra Leone, 3 Sustainable Health Systems Sierra Leone, Freetown, Sierra Leone, 4 Global Action For Fungal Infections, Geneva, Switzerland, 5 Department of Medicine, College of Medicine, University of Ibadan, Ibadan, Nigeria, 6 Department of Medicine, University College Hospital, Ibadan, Nigeria, 7 Division of Clinical Epidemiology, University Hospital Basel, Basel, Switzerland, 8 Clinical Research Unit, Department of Medicine, Swiss Tropical and Public Health Institute, Basel, Switzerland, 9 University of Basel, Basel, Switzerland, 10 Centre for Multidisciplinary Research and Innovation, Abuja, Nigeria, 11 Department of Medicine, Case Western Reserve University School of Medicine, Cleveland, Ohio, United States of America, 12 Division of Infectious Diseases and HIV Medicine, University Hospitals Cleveland Medical Center, Cleveland, Ohio, United States of America, 13 Johns Hopkins Bloomberg School of Public Health, Baltimore, Maryland, United States of America, 14 Manchester Fungal Infection Group, University of Manchester, Manchester Academic Health Science Centre, Manchester, United Kingdom

* lakoh2009@gmial.com

## Abstract

### Background

In the World Health Organization Global Tuberculosis (TB) Report 2022, 37% of pulmonary TB patients were clinically diagnosed and thus many people were treated for TB without evidence of the disease. Probably the most common TB misdiagnosis is chronic pulmonary aspergillosis (CPA). In this study, we aimed to assess the prevalence and predictors of *Aspergillus* seropositivity and CPA in patients with chronic respiratory symptoms in an urban tertiary hospital in Sierra Leone.

### Methodology/principal findings

We used a cross-sectional study design to recruit adults (≥18 years) from the Chest Clinic of Connaught Hospital, Freetown between November 2021 and July 2022. *Aspergillus* antibody was detected using LDBio *Aspergillus* IgM/IgG. Logistic regression was performed to assess the independent predictors of *Aspergillus* seropositivity and CPA. Of the 197 patients with chronic respiratory symptoms, 147 (74.6%) were male. Mean age was 47.1 ± 16.4 years. More than half (104, 52.8%) had been diagnosed with TB in the past, while 53 (26.9%) were on TB treatment at the time of recruitment. Fifty-two (26.4%) patients were

**Data Availability Statement:** All relevant data are within the paper and its Supporting Information files.

**Funding:** The project received materials and technical support from the Global Action For Fungal Infections (GAFFI) in designing the study and preparing the manuscript. The funder had no additional role. Data collection was supported financially by UK Research and Innovation as part of the Global Challenges Research Fund, Grant Number ES/P010873/1, which had no role in study design, data collection and analysis, decision to publish, or preparation of the manuscript.

**Competing interests:** The authors have declared that no competing interests exist.

HIV positive, 41 (20.8%) were seropositive for *Aspergillus* and 23 (11.6%) had CPA, 2 (3.8%) with current TB and 18 (17.3%) with past TB. Common radiologic abnormalities reported were localized fibrotic changes 62 (31.5%), consolidation 54 (27.4%), infiltrates 46 (23.4%), hilar adenopathy 40 (20.3%) and pleural effusion 35 (17.85) and thickening 23 (11.7%). Common symptoms were weight loss 144 (73.1%), cough 135 (68.5%), fever 117 (59.4%) and dyspnea 90 (45.7%). Current or past TB infection {aOR 3.52, 95% CI (1.46, 8.97); p = 0.005} was an independent predictor of *Aspergillus* seropositivity and CPA.

## Conclusions/significance

We report a high prevalence of *Aspergillus* antibody seropositivity and CPA, underscoring the need to integrate the prevention and management of pulmonary fungal infections with TB services and asthma care in order to reduce unnecessary morbidity and mortality.

## Author summary

Chronic pulmonary aspergillosis (CPA) is a common cause of chronic lung disease. It mimics tuberculosis (TB), and can occur during or after TB treatment, mainly in patients with lung cavities. Since nearly 40% of TB cases worldwide are undiagnosed microbiologically, CPA may be the most common cause of symptoms in patients treated for TB without a microbiological diagnosis. Understanding the burden of CPA using the *Aspergillus* antibody test is an important initial step in addressing this persistent and chronic neglected disease in low-resource settings and presents an opportunity for healthcare workers to acquire the skills needed to reduce unnecessary CPA-related mortality. This study assessed the prevalence of *Aspergillus* seropositivity and CPA and found that 20.8% of patients were positive for *Aspergillus* antibodies and 11.6% had CPA. Common symptoms were cough, weight loss, difficulty breathing and fever and TB was an independent predictor of *Aspergillus* seropositivity and CPA.

## Background

The prevention and control of fungal diseases like chronic pulmonary aspergillosis (CPA) remains a major challenge, especially in resource-poor countries [1,2]. Estimates of global post-TB prevalence of CPA based on 2005 TB data suggest that approximately 1.2 million people are affected [3]. This fact demonstrates the trajectory of the increasing burden of CPA and supports recent estimates of 3 million global cases of CPA [4].

*Aspergillus* can coexist with or complicate tuberculosis (TB) and other chronic lung diseases and thus may pose a significant challenge to their management [3]. In the World Health Organization (WHO) Global TB Report 2022, approximately 37% of pulmonary TB patients were clinically diagnosed (i.e., were smear/Xpert negative or not done) [5]. Thus, many people were treated for TB without evidence of the disease. Probably the most common TB misdiagnosis is CPA, which causes considerable morbidity and unnecessary mortality. CPA can mimic TB and can occur during or after TB treatment, mainly in those with lung cavities [4].

CPA is a global health problem that primarily affects low- and middle-income countries. A study in Iran found that 6% of people had CPA at the time of their first TB episode within 12 months of diagnosis [6]. A recent study in Indonesia found that 13% of patients developed

CPA after 6 months of TB treatment [7]. Recent data from India and Vietnam found that more than 50% of TB patients with new symptoms had CPA [8,9].

In sub-Saharan Africa, there are no comprehensive estimates of the number of misdiagnosed TB when the diagnosis is actually CPA. A prospective longitudinal study in northern Uganda found that 6.5% of people with cavities each year developed CPA 2–7 years after TB, compared with 0.2% of people without cavities [10]. A study from Nigeria found that 8.7% of the smear/Xpert-negative population had CPA [11]. In Ghana, 9.7% of patients with presumed TB had CPA [12]. We recently estimated that 6000 people in Sierra Leone suffer from CPA [13].

Sierra Leone is one of the 30 countries with the highest TB burden in the world, with 24,000 cases (298 TB cases per 100,000 population) in 2021 [3]. In our recent studies, nearly 50% of patients treated for TB at the national referral hospital in Sierra Leone were clinically diagnosed [14,15]. Despite this and the fact that CPA is probably the main cause of their symptoms, there is limited data on the burden of this disease among patients with chronic respiratory symptoms in Sierra Leone owing to the limited human resource and diagnostic capacity for the detection, management and prevention of this fungal infections. The study aimed to assess prevalence and predictors of *Aspergillus* seropositivity and CPA in patients with chronic respiratory symptoms in an urban tertiary hospital in Sierra Leone in order to determine the local needs for training of healthcare workers on the detection and management of CPA, with the overall aim of improving health outcomes of people with CPA in Sierra Leone.

## Methods

### Ethics statement

Ethics approval was obtained from the Sierra Leone Ethics and Scientific Review Committee (SLESRC) of the Ministry of Health and Sanitation, Government of Sierra Leone in accordance with the relevant guidelines and regulations and declaration of Helsinki. Approval to conduct this study was granted by SLESRC, dated 16[th] June 2021. Written informed consent was obtained from individual participants before enrolling in the study.

### Study design

We used a cross-sectional study design to collect data from adult patients aged 18 years or older from the Chest Clinic of Connaught Hospital, Freetown, Sierra Leone.

### Study setting

Sierra Leone is divided into five geographical regions, including the Western Area. The Western Area is the most populous region of Sierra Leone and includes Freetown (the capital city). According to the 2015 population census, Sierra Leone has a total population of 7 million, of which 22% (1.5 million) live in the Western Area [16].

Of the 25 public hospitals in Sierra Leone, 10 provide tertiary care. The study was conducted at Connaught Hospital, Sierra Leone's main tertiary hospital in Western Area with a capacity of 300 beds. Connaught Hospital's Chest Clinic has the largest number of TB patients in the country.

### Sampling method

Though a minimal sample size of 118 patients was obtained with Fisher's formula using prevalence of 8.7% of CPA in Nigeria [11], with a coefficient interval of 95% and a margin of error of 0.05, a total of 197 patients were recruited.

Adults ($\geq$18 years) with respiratory symptoms were recruited sequentially from the first week of November 2021 to reach a sample size of 197 patients (approximately 30 weeks). Demographic and clinical information were collected from patients using a standardized data collection form and cross-checked with clinical records. Symptoms and chest radiograph findings were recorded using a diagnostic algorithm published in 2018 [17]. The presumed pneumonia or bronchitis cases were inferred from antibiotic use in this patient cohort.

## Laboratory procedure

Blood samples were aseptically collected into EDTA test tubes and immediately centrifuged at 4000 revolutions per minutes for 5 minutes to generate plasma. Separated plasma were then analyzed using *Aspergillus* IgM/IgG (LDBIO Diagnostics, Lyon, France) point-of-care immunochromatographic (ICT) lateral flow assay [18]. The LDBio *Aspergillus* IgM/IgG ICT test has a sensitivity and specificity of 88.9% and 96.3%, respectively. Similar to other point-of-care tests, it has been added to the 2021 WHO Model of Essential Diagnostic List (EDL) [18,19].

TB was diagnosed using the Xpert *Mycobacterium tuberculosis* (MTB)/RIF assay (Cepheid) with compatible clinical features and chest radiograph findings [20]. HIV test was performed using Determine and SD Bioline HIV-1/2 3.0 test kits (Standard Diagnostics Inc) in the National HIV Testing Algorithm [21]. CD4 cell count was determined using the Alere Pima™ Analyzer (Abbott), a point-of-care testing platform validated in resource-limited settings [22].

## Definitions of Aspergillus seropositivity and CPA

Patients with chronic respiratory symptoms such as weight loss, cough, and/or hemoptysis persisting for more than 3 months and positive for LDBIO *Aspergillus* IgG were considered to have *Aspergillus* seropositivity regardless of their radiological findings.

The Global Action For Fungal Infections (GAFFI) and its international experts established a case definition for CPA in 2018 to support research and clinical and public health practices in low- and middle-income countries [23]. The GAFFI panel defined CPA as an illness lasting more than three months with the presence of all the typical symptoms (weight loss, persistent cough, and/or hemoptysis) and chest radiograph showing progressive cavitary infiltrates and/or a fungal ball and/or peri-cavitary fibrosis or infiltrates or pleural thickening in patients with a positive *Aspergillus* IgG serology [23]. Chest radiographs were read and reported by Radiologists and further reviewed by the research team using guidance provided by Leading the International Fungal Education (LIFE) Worldwide (available at: https://www.youtube.com/watch?v=zuYoLW-n_2w).

We applied the GAFFI case definition to categorized confirmed CPA and the remaining patients with only two of the key features as probable CPA. In keeping with standard practice, patients with probable and confirmed CPA were given oral itraconazole 400 mg per day for a minimum of 6 months.

## Data management and analysis

The data was collected using the Epicollect data platform, pooled into an Excel sheet and clean, coded and exported to SPSS 21.0 (Armonk, NY: IBM Corp) for analysis. The data were summarized using frequencies and measures of central tendency. Normally distributed numerical variables were summarized using mean and standard deviation (SD). Categorical variables were summarized with frequencies and proportions. Demographic, clinical and radiographic characteristics were compared between patients who are seropositive for *Aspergillus* or had CPA using chi-square or Fisher's exact test, as appropriate. Logistic regression analysis was used to identify potential predictors of *Aspergillus* seropositivity and CPA. Only

**Table 1. Socio-demographic and clinical characteristics of study population.**

| Characteristics | N (%) |
|---|---|
| **Gender** | |
| Male | 147 (74.6) |
| Female | 50 (25.4) |
| **Age,** *years* | |
| Mean ± SD | 47.1 ± 16.4 |
| < 30 | 30 (15.2) |
| 30–39 | 44 (22.3) |
| 40–49 | 36 (18.3) |
| 50–59 | 39 (19.8) |
| 60+ | 48 (24.4) |
| **Body mass index,** kg/m$^2$ | |
| Mean ± SD | 17.4 ± 3.7 |
| < 18.5 (underweight) | 124 (62.9) |
| 18.5–24.9 (normal) | 70 (35.5) |
| >25 (overweight) | 3 (1.5) |
| **Relationship status** | |
| Single | 63 (32.0) |
| Married | 112 (56.9) |
| Divorced, separated or widowed | 22 (11.2) |
| **Highest education attained** | |
| None | 46 (23.4) |
| Primary | 24 (12.2) |
| Secondary | 85 (43.1) |
| Tertiary | 42 (21.3) |
| **Smoking** | |
| Yes | 81 (41.1) |
| No | 116 (58.9) |
| **Alcohol use** | |
| Yes | 77 (39.1) |
| No | 120 (60.9) |
| **Drug use** | |
| Yes | 8 (4.1) |
| No | 189 (95.9) |
| **HIV status** | |
| Positive | 52 (26.4) |
| Negative | 132 (67.0) |
| Unknown | 13 (6.6) |
| **CD4 count,** cells/mm$^3$ (n = 52) | |
| Mean ± SD | 161 ± 93 |
| < 100 | 11 (5.6) |
| 100–199 | 35 (17.8) |
| ≥200 | 6 (3.0) |
| **Current Tuberculosis** | |
| Yes | 53 (26.9) |
| No | 144 (73.1) |
| **Past Tuberculosis** | |
| Yes | 104 (52.8) |

(*Continued*)

**Table 1.** (Continued)

| Characteristics | N (%) |
|---|---|
| No | 93 (47.2) |
| **Chronic lung diseases** | |
| Asthma | 8 (4.1) |
| COPD | 8 (4.1) |
| **Presumed bacterial pneumonia or bronchitis** | |
| Yes | 129 (65.5) |
| No | 68 (34.5) |
| **Duration of acute respiratory symptoms,** *days* | |
| Median (IQR) | 8 (3–12) |
| < 7 days | 95 (48.2) |
| ≥ 7 days | 102 (51.8) |
| *Aspergillus* **seropositivity** | |
| Positive | 41 (20.8) |
| Negative | 156 (79.2) |
| **Chronic pulmonary aspergillosis (CPA)** | |
| Yes | 23(11.6%) |
| No | 174(88.4%) |

variables which attained p < 0.20 in univariate analysis were included in the final multivariable regression model. The level of significance for all tests was set at *p< 0.05* with a 95% confidence interval.

## Results

### Sociodemographic characteristics and associated disease

Of the 197 patients with chronic respiratory symptoms, 147 (74.6%) were male. The mean age is 47.1 ± 16.4 years. Many patients were married 112 (56.9%) and had secondary education as their highest education level 85 (43.1%). The mean BMI was 17.4 ± 3.7 kg/m$^2$. More than half (104, 52.8%) had been diagnosed with TB in the past, while 53 (26.9%) were on anti-TB treatment at the time of recruitment. Fifty-two (26.4%) patients were HIV positive, 129 (65.5%) had recent or current pneumonia or bronchitis, and 8 (4.1%) each had chronic obstructive pulmonary disease (COPD) or asthma. This method is prone to error but not likely to differ radically from the true number of pneumonia cases given the usual pattern of clinical practice. Pneumonia/bronchitis was not verified by radiology or microbiology testing. Forty-one (20.8%) patients were positive for *Aspergillus* IgG/IgM antibody (Tables 1 and 2).

### Radiologic findings

Chest radiographs were available for all 197 patients with chronic respiratory symptoms, many of whom showed compatible features with CPA. Fig 1A–1C provide examples of typical x-ray changes in patients with chronic respiratory symptoms. Xray changes reported were bilateral in 100 (50.8%) patients. Common abnormalities reported were localized fibrotic changes in 62 (31.5%), consolidation in 54 (27.4), infiltrates in 46 (23.4%), hilar adenopathy in 40 (20.3%), pleural effusion in 35 (17.9%) and pleural thickening in 23 (11.7%) (Table 3).

   *Aspergillus* antibody seropositivity (n = 41) was associated with extensive lung fibrosis (36.6% vs. 20.3%) and peri-cavitary fibrosis/thickening (12.2% vs. 7.1%) (p < 0.001), pleural thickening (14.6% vs. 10.9%) and cavitary changes with or without intracavitary fungal ball

**Table 2. Aspergillus IgM/IgM or CPA by sociodemographic and clinical factors.**

| Characteristics | Aspergillus IgM/IgG | | | | CPA | | | |
|---|---|---|---|---|---|---|---|---|
| | All (%) | Positive (%) | Negative (%) | p-Value | All (%) | Yes (%) | No (%) | p-Value |
| **Gender** | | | | | | | | |
| Male | 147 (74.6) | 289 (70.7) | 118 (75.6) | 0.520 | 147 (74.6) | 17 (73.9) | 130 (74.7) | 0.934 |
| Female | 50 (25.4) | 12 (29.3) | 38 (24.4) | | 50 (25.4) | 6 (26.1) | 44 (25.3) | |
| **Age,** *years* | | | | | | | | |
| < 30 | 30 (15.2) | 5 (12.2) | 25 (16.0) | 0.890 | 30 (15.2) | 2 (8.7) | 28 (16.1) | 0.675 |
| 30–39 | 44 (22.3) | 10 (24.4) | 34 (21.8) | | 44 (22.3) | 6 (26.1) | 38 (21.8) | |
| 40–49 | 36 (18.3) | 7 (17.1) | 29 (18.6) | | 36 (18.3) | 6 (26.1) | 30 (17.2) | |
| 50–59 | 39 (19.8) | 10 (24.4) | 29 (18.6) | | 39 (19.8) | 5 (21.7) | 34 (19.5) | |
| 60+ | 48 (24.4) | 9 (22.0) | 39 (25.0) | | 48 (24.4) | 4 (17.4) | 44 (25.3) | |
| **Body mass index**, kg/m$^2$ | | | | | | | | |
| < 18.5 (underweight) | 124 (62.9) | 30 (73.2) | 94 (60.3) | 0.253 | 124 (62.9) | 16 (69.6) | 108 (62.1) | 0.681 |
| 18.5–24.9 (normal) | 70 (35.5) | 11 (26.8) | 59 (37.8) | | 70 (35.5) | 7 (30.4) | 63 (36.2) | |
| >25 (overweight) | 3 (1.5) | - | 3 (1.5) | | 3 (1.5) | - | 3 (1.7) | |
| **Relationship status** | | | | | | | | |
| Single | 63 (32.0) | 12 (29.3) | 51 (32.7) | 0.908 | 63 (32.0) | 4 (17.4) | 59 (33.9) | 0.279 |
| Married | 112 (56.9) | 24 (58.5) | 88 (56.4) | | 112 (56.9) | 16 (69.6) | 96 (55.2) | |
| Divorced, separated or widowed | 22 (11.1) | 5 (12.2) | 17 (10.9) | | 22 (11.1) | 3 (13.0) | 19 (10.9) | |
| **Highest education attained** | | | | | | | | |
| None | 46 (23.4) | 9 (22.0) | 37 (23.7) | 0.835 | 46 (23.4) | 5 (21.7) | 41 (23.6) | 0.456 |
| Primary | 24 (12.2) | 5 (12.2) | 19 (12.2) | | 24 (12.2) | 1 (4.3) | 23 (13.2) | |
| Secondary | 85 (43.1) | 20 (48.8) | 65 (41.7) | | 85 (43.1) | 13 (56.5) | 72 (41.4) | |
| Tertiary | 42 (21.3) | 7 (17.1) | 35 (22.4) | | 42 (21.3) | 4 (17.4) | 38 (21.8) | |
| **Smoking** | | | | | | | | |
| Yes | 81 (41.1) | 12 (29.3) | 69 (44.2) | 0.083 | 81 (41.1) | 6 (26.1) | 75 (43.1) | 0.119 |
| No | 116 (58.9) | 29 (70.7) | 87 (55.8) | | 116 (58.9) | 17 (73.9) | 99 (56.9) | |
| **Alcohol use** | | | | | | | | |
| Yes | 77 (39.1) | 14 (34.1) | 63 (40.4) | 0.466 | 77 (39.1) | 10 (43.5) | 67 (38.5) | 0.646 |
| No | 120 (60.9) | 27 (65.9) | 93 (59.6) | | 120 (60.9) | 13 (56.5) | 107 (61.5) | |
| **Drug use** | | | | | | | | |
| Yes | 8 (4.1) | 1 (2.4) | 7 (4.5) | 0.554 | 8 (4.1) | 1 (4.3) | 7 (4.0) | 0.941 |
| No | 189 (95.9) | 40 (97.6) | 149 (95.5) | | 189 (95.9) | 22 (95.7) | 167 (96.0) | |
| **HIV status (n = 184)** | | | | | | | | |
| Positive | 52 (28.3) | 5 (13.2) | 47 (32.2) | **0.020** | 52 (28.3) | 2 (9.1) | 50 (30.9) | **0.033** |
| Negative | 132 (71.7) | 33 (86.8) | 99 (67.8) | | 132 (71.7) | 20 (90.9) | 112 (69.1) | |
| **CD4 count,** cells/mm$^3$ (n = 52) | | | | | | | | |
| < 200 | 44 (84.6) | 4 980.0) | 40 (85.1) | 0.764 | 46 (23.4) | 2 (8.7) | 44 (25.3) | 0.117 |
| ≥200 | 8 (15.4) | 1 (20.0) | 7 (14.9) | | | | | |
| **Current Tuberculosis** | | | | | | | | |
| Yes | 53 (26.9) | 3 (7.3) | 50 (32.1) | **0.001** | 53 (26.9) | 2 (8.7) | 51 (29.3) | **0.036** |
| No | 144 (73.1) | 38 (92.7) | 106 (67.9) | | 144 (73.1) | 21 (91.3) | 123 (70.7) | |
| **Past Tuberculosis** | | | | | | | | |
| Yes | 104 (52.8) | 31 975.6) | 73 (46.8) | **0.001** | 104 (52.8) | 18 (78.3) | 86 (49.4) | **0.009** |
| No | 93 (47.2) | 10 (24.4) | 83 (53.2) | | 93 (47.2) | 5 (21.7) | 88 (50.6) | |
| **Xpert smear** | | | | | | | | |
| Positive | 67 (34.0) | 20 (48.8) | 47 (30.1) | **0.002** | 67 (34.0) | 10 (43.5) | 57 (32.8) | **0.025** |
| Negative | 38 (19.3) | 12 (29.3) | 26 (16.7) | | 38 (19.3) | 8 (34.8) | 30 (17.2) | |

*(Continued)*

**Table 2.** (Continued)

| Characteristics | Aspergillus IgM/IgG | | | | CPA | | | |
|---|---|---|---|---|---|---|---|---|
| | All (%) | Positive (%) | Negative (%) | p-Value | All (%) | Yes (%) | No (%) | p-Value |
| Not available | 92 (46.7) | 9 (22.0) | 83 (53.2) | | 92 (46.7) | 5 (21.7) | 87 (50.0) | |
| **Asthma** | | | | | | | | |
| Yes | 8 (4.1) | 4 (9.8) | 4 (2.6) | 0.060 | 8 (4.1) | 1 (4.3) | 7 (4.0) | 1.000 |
| No | 189 (95.9) | 37 (90.2) | 152 (97.4) | | 189 (95.9) | 22 (95.7) | 167 (96.0) | |
| **COPD** | | | | | | | | |
| Yes | 8 (4.1) | 1 (2.4) | 7 (4.5) | 1.000 | 8 (4.1) | - | 8 (4.6) | 0.600 |
| No | 189 (95.9) | 40 (97.6) | 149 (95.5) | | 189 (95.9) | 23 (100) | 166 (95.4) | |
| **Presumed bacterial pneumonia or bronchitis** | | | | | | | | |
| Yes | 129 (65.5) | 23 (56.1) | 106 (67.9) | 0.156 | 129 (65.5) | | | |
| No | 68 (34.5) | 18 (43.9) | 50 (32.1) | | 68 (34.5) | | | |
| **Symptoms** | | | | | | | | |
| Fever | 117 (59.4) | 21 (51.2) | 96 (61.5) | 0.231 | 117 (59.4) | 13 (56.5) | 104 (59.8) | 0.766 |
| Cough | 135 (68.5) | 27 (65.9) | 108 (69.2) | 0.679 | 135 (68.5) | 14 (60.9) | 121 (69.5) | 0.400 |
| Weight loss | 144 (73.1) | 25 (61.0) | 119 (76.3) | **0.049** | 144 (73.1) | 12 (52.2) | 132 (75.90) | **0.016** |
| Dyspnea | 90 (45.7) | 18 (43.9) | 72 (46.2) | 0.797 | 90 (45.7) | 10 (43.5) | 80 (46.0) | 0.821 |
| Pleuritic chest pain | 46 (23.4) | 8 (19.5) | 38 (24.4) | 0.545 | 46 (23.4) | 5 (21.7) | 104 (23.6) | 0.846 |
| Night sweats | 44 (22.3) | 5 (12.2) | 39 (25.0) | 0.080 | 44 (23.3) | 2 (8.7) | 42 (24.1) | 0.095 |
| Hemoptysis | 35 (17.8) | 9 (22.0) | 26 (16.7) | 0.431 | 35 (17.8) | 4 (17.4) | 31 (17.8) | 0.960 |

(14.6% vs. 11.7%) (Table 2). In contrast, consolidation is more frequent in those with a negative *Aspergillus* IgG/IgM (32.7% vs. 7.3%) (p = 0.001) and hilar adenopathy (24.4% vs. 4.9%) (p = 0.06). CPA (n = 23) was more frequent in those with cavitary changes (26.1 vs. 10.9) (p = 0.04) as well as both pleural effusion and pleural thickening (p = 0.008) and extensive lung fibrosis and peri-cavitary fibrosis/thickening (p = 0.002) (Table 3).

## Symptoms

Respiratory symptoms of the 197 patients are shown in Table 2. Common symptoms were weight loss 144 (73.1%), cough 135 (68.5%), fever 117 (59.4%) and dyspnea 90 (45.7%). Weight loss was significantly less frequent in those with *Aspergillus* antibody seropositivity (60.9% vs 76.3%) and CPA (52.2% vs. 75.9%) (Table 2). No other symptoms distinguished CPA from other conditions.

## Prevalence of CPA

Amongst the *Aspergillus* seropositive patients, 23 (11.6%) had CPA (Table 1). Of the 23 CPA patients, 10 (43.5%) had detectable *Mycobacterium tuberculosis* (MTB) on Xpert MTB/Rif, 18 (78.3%) had received TB treatment in the past, 2 (8.7%) had dual HIV/CPA infections (Table 2).

Amongst the 53 patients with current TB, 3 (4.1%) and 2 (3.8%) were *Aspergillus* seropositive or had CPA respectively (Table 2). Of those with past TB (n = 104), 31 (29.8%) had a positive *Aspergillus* antibody result and 18 (17.3%) had CPA. The 92 patients without TB had completed TB treatment and were therefore not eligible for a repeat TB test by national standards (n = 8), recruited at a time when there were disruptions in the supply of GeneXpert cartridges (n = 72) or unable to produce sputum for Xpert test (n = 12).

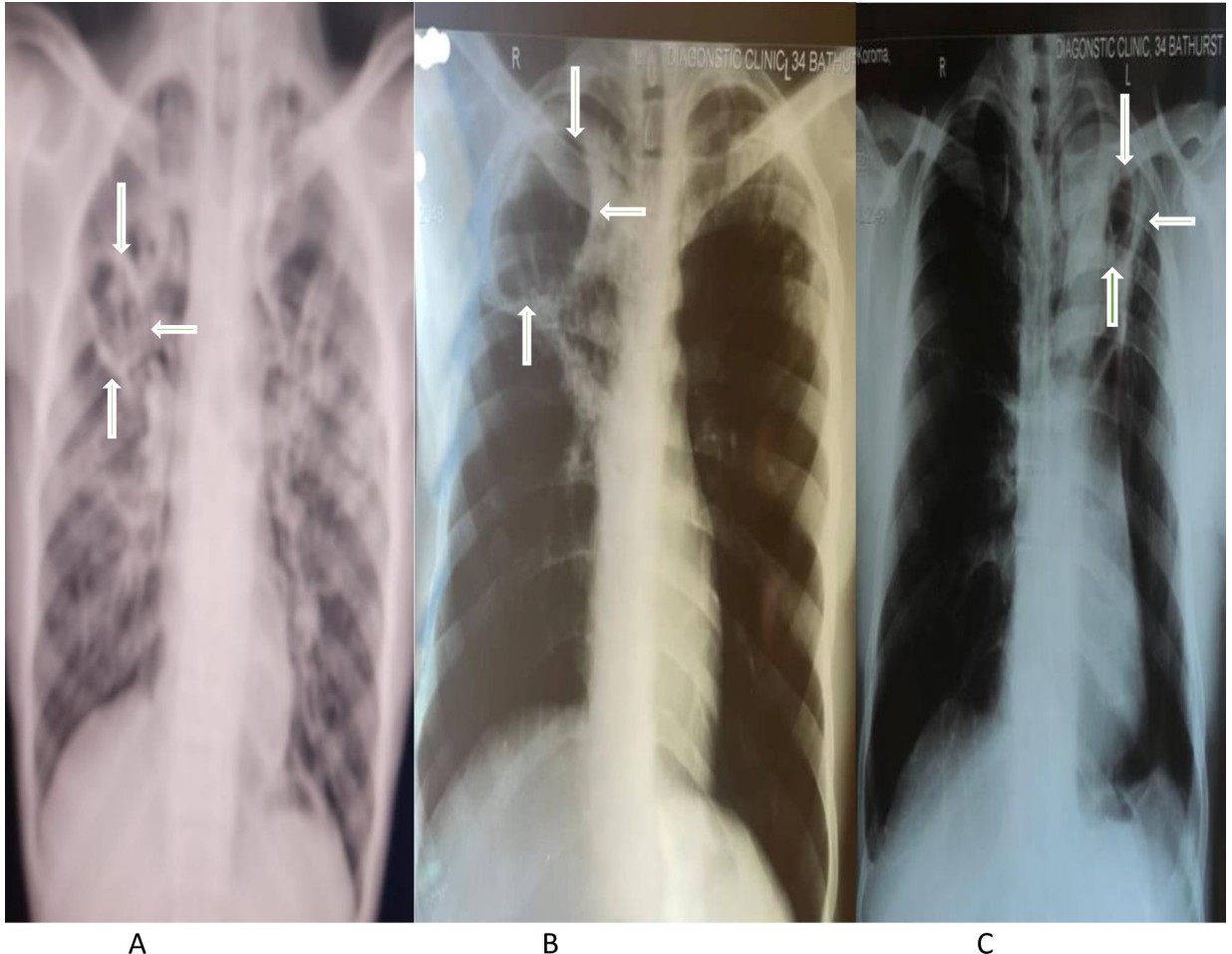

**Fig 1.** A-C: Images of chest radiographs of patients with a positive *Aspergillus* antibody.

### Predictors of *Aspergillus* seropositivity and CPA

In multivariable analyses, past or current TB infection {aOR 3.52, 95% CI (1.46, 8.97); p = 0.005} and low BMI {aOR 0.2.56, 95%CI (1.01, 6.49); p = 0.047} were independent predictors of a positive *Aspergillus* seropositivity. People with positive HIV test {aOR 0.25, 95% CI (0.09, 0.74); p = 0.012} and those smoking cigarette {aOR 0.41, 95% CI (1.17, 0.74); p = 0.044} were less likely to have a positive *Aspergillus* IgM/IgG (Table 4). In those with HIV infection, CD4 count (above or below 200x10$^6$) was not linked to *Aspergillus* seropositivity or CPA. Asthma patients were 7.99 more likely to have a positive *Aspergillus* seropositivity {aOR 7.99, 955CI (1.55,55.42)}. There is no association between *Aspergillus* seropositivity and COPD or pneumonia/bronchitis (Table 4). Likewise, asthma, COPD and pneumonia/bronchitis were not linked to CPA. The presence of past or current TB infection {aOR 3.71, 95% CI (1.23, 11.24); p = 0.008} was the only independent predictor of CPA on multivariable analysis (Table 5).

### Discussion

We analyzed the prevalence and predictors of *Aspergillus* seropositivity and CPA in adult patients with chronic respiratory symptoms in a national referral hospital in Sierra Leone.

**Table 3. Chest radiographic findings in the patients.**

| Imaging findings | Aspergillus IgG/IgM | | | | Chronic pulmonary aspergillosis | | | |
|---|---|---|---|---|---|---|---|---|
| | All (%) | Positive (%) | Negative (%) | p-Value | All (%) | Yes (%) | No (%) | p-Value |
| **Laterality of findings** | | | | | | | | |
| Unilateral | 97 (49.2) | 18 (42.9) | 79 (49.2) | 0.442 | 97 (49.20) | 14 (60.9) | 86 (49.4) | 0.302 |
| Bilateral | 100 (50.8) | 23 (56.1) | 77 (49.4) | | 100 (50.8) | 9 (39.1) | 88 (50.6) | |
| **Infiltrates** | | | | | | | | |
| Yes | 46 (23.4) | 10 (24.4) | 36 (23.1) | 0.860 | 46 (23.4) | 10 (43.5) | 36 (20.7) | **0.015** |
| No | 151 (76.6) | 31 (75.6) | 120 (76.9) | | 151 (76.6) | 13 (56.5) | 138 (79.3) | |
| **Consolidation** | | | | | | | | |
| Yes | 54 (27.4) | 3 (7.3) | 51 (32.7) | **0.001** | 54 (27.4) | 9 (39.1) | 45 (25.9) | 0.180 |
| No | 143 (72.6) | 38 (92.7) | 105 (67.3) | | 143 (72.6) | 14 (60.9) | 129 (74.1) | |
| **Hilar adenopathy** | | | | | | | | |
| Yes | 40 (20.3) | 2 (4.9) | 38 (24.4) | **0.006** | 40 (20.3) | 6 (26.1) | 34 (19.5) | 0.463 |
| No | 157 (79.7) | 39 (95.1) | 156 (75.6) | | 157 (79.7) | 17 (73.9) | 140 (80.5) | |
| **Cavitary changes** | | | | | | | | |
| Yes | 25 (12.7) | 6 (14.6) | 19 (12.2) | 0.674 | 25 (12.7) | 6 (26.1) | 19 (10.9) | **0.040** |
| No | 172 (87.3) | 35 (85.4) | 137 (87.8) | | 172 (87.3) | 17 (73.9) | 155 (89.1) | |
| **Pleural changes** | | | | | | | | |
| None | 139 (70.6) | 23 (56.10 | 116 (74.4) | 0.056 | 139 (70.6) | 10 (43.5) | 129 (74.1) | **0.008** |
| Pleural effusion | 35 (17.8) | 12 (29.3) | 23 (14.7) | | 35 (17.8) | 7 (30.4) | 28 (16.1) | |
| Pleural thickening | 23 (11.7) | 6 (14.6) | 17 (10.9) | | 23 (11.7) | 6 (26.1) | 17 (9.8) | |
| **Fibrotic changes** | | | | | | | | |
| None | 79 (40.1) | 4 (9.8) | 25 (16.0) | < **0.001** | 79 (40.1) | 1 (4.3) | 78 (44.8) | **0.002** |
| Extensive | 40 (20.3) | 15 (36.6) | 45 (28.8) | | 40 (20.3) | 7 (30.4) | 33 (19.0) | |
| Localized | 62 (31.5) | 17 (41.5) | 75 (48.1) | | 62 (31.5) | 11 (47.8) | 51 (29.3) | |
| Para-cavitary | 16 (8.1) | 5 (12.2) | 11 (7.1) | | 16 (8.1) | 4 (17.4) | 12 (6.9) | |

Amongst the 197 patients with chronic respiratory symptoms, 20.8% had a positive serum *Aspergillus* antibody assay, higher than the serological evidence of the 19.5% reported among TB patients with persistent pulmonary symptoms in Kenya [24]. *Aspergillus* IgG in combination with compatible clinical symptoms and radiologic findings has become the preferred approach for diagnosing CPA [25]. Using the case definition developed by GAFFI [23], we reported a CPA prevalence of 11.6%, split between those being treated for TB (3.8%) and those with a past history of TB (17.3%). A lower prevalence of CPA was reported in Ghanaian patients with suspected TB (9.7%) and Nigerian patients with smear-negative TB (8.7%) [11,12].

The higher prevalence of CPA in our study could be attributed to the additional culture method applied in the detection of CPA in the Nigerian and Ghanian studies. Evidence is accumulating that false-positive results of *Aspergillus* IgG exist, because a positive *Aspergillus* antibody is not specific to CPA, and can represent several other *Aspergillus*-related disorders, including *Aspergillus* rhinosinusitis, allergic bronchopulmonary aspergillosis, *Aspergillus* bronchitis and subacute invasive aspergillosis. We partly address this by categorizing the remaining patients with only two of three key features as probable CPA [23]. Regardless of the category, the high burden of *Aspergillus* seropositivity reported in our study reflect on our earlier call to improve on the diagnostic capacity of fungal infections in Africa and underscores the need to initiate routine screening among patients with chronic respiratory symptoms and provide timely and appropriate anti-fungal treatment to those with CPA [26,27].

**Table 4. Factors associated with *Aspergillus* seropositivity.**

| Characteristics | *Aspergillus* seropositivity | | Univariate | | Multivariate | |
|---|---|---|---|---|---|---|
| | Yes | No | Crude OR (95% CI) | p-Value | Adjusted OR (95% CI) | p-Value |
| **Gender** | | | | | | |
| Male | 29 (70.9) | 118 (75.6) | 0.78 (0.36–1.67) | 0.520 | | |
| Female | 12 (29.3) | 38 (24.4) | Ref | | | |
| **Age,** *years* | | | | | | |
| < 45 | 20 (48.8) | 71 (45.5) | 1.14 (0.57–2.27) | 0.709 | | |
| ≥45 | 21 (51.2) | 85 (54.5) | Ref | | | |
| **Body mass index**, kg/m$^2$ | | | | | | |
| < 18.5 | 30 (73.2) | 94 (60.3) | 1.80 (0.84–3.85) | 0.128 | 2.70 (1.05–6.93) | **0.039** |
| ≥18.5 | 11 (26.8) | 62 (39.7) | Ref | | Ref | |
| **Relationship status** | | | | | | |
| Single | 12 (29.3) | 51 (32.7) | 0.85 (0.40–1.81) | 0.676 | | |
| Married/others | 29 (70.7) | 105 (67.3) | Ref | | | |
| **Highest education attained** | | | | | | |
| None | 9 (22.0) | 37(23.7) | 0.90 (0.40–2.07) | 0.812 | | |
| Primary or higher | 32 (78.0) | 119 (76.3) | Ref | | | |
| **Smoking** | | | | | | |
| Yes | 12 (29.3) | 69 (44.2) | 0.52 (0.25–1.10) | 0.083 | 0.44 (0.18–1.08) | 0.074 |
| No | 29 (70.7) | 87 (55.8) | Ref | | Ref | |
| **Alcohol use** | | | | | | |
| Yes | 14 (34.1) | 63 (40.4) | 0.77 (0.37–1.57) | 0.466 | | |
| No | 27 (65.9) | 93 (59.6) | Ref | | | |
| **Drug use** | | | | | | |
| Yes | 1 (2.4) | 7 (4.5) | 0.53 (0.06–4.45) | 1.000 | | |
| No | 40 (97.6) | 149 (95.5) | Ref | | | |
| **HIV status** | | | | | | |
| Positive | 5 (13.2) | 47 (32.2) | 0.32 (0.18–0.87) | **0.020** | 0.23 (0.09–0.60) | **0.012** |
| Negative | 33 (86.8) | 99 (67.8) | Ref | | Ref | |
| **CD4 count**, cells/mm$^3$ | | | | | | |
| < 200 | 4 (80.0) | 42 (89.4) | 0.48 (0.04–5.14) | 0.533 | | |
| ≥200 | 1 (20.0) | 5 (10.6) | Ref | | | |
| **Past or current tuberculosis** | | | | | | |
| Yes | 32 (78.0) | 73 (46.8) | 4.04 (1.81–9.03) | **< 0.001** | 4.63 (1.41–15.24) | **0.005** |
| No | 9 (22.0) | 83 (53.2) | Ref | | Ref | |
| **Asthma** | | | | | | |
| Yes | 4 (9.8) | 4 (2.6) | 4.11 (0.98–17.20) | 0.060 | 7.99 (1.15–55.42) | **0.036** |
| No | 37 (90.2) | 152 (97.4) | Ref | | Ref | |
| **COPD** | | | | | | |
| Yes | 1 (2.4) | 7 (4.5) | 0.53 (0.06–4.45) | 1.000 | | |
| No | 40 (97.6) | 149 (95.5) | Ref | | | |
| **Presumed bacterial pneumonia or bronchitis** | | | | | | |
| Yes | 23 (56.1) | 106 (67.9) | 0.60 (0.30–1.22) | 0.156 | 0.53 (0.22–1.27) | 0.155 |
| No | 18 (43.9) | 50 (32.1) | Ref | | Ref | |
| **Duration of respiratory symptoms,** *days* | | | | | | |
| < 7 days | 23 (56.1) | 72 (46.2) | 1.49 (0.75–2.98) | 0.257 | | |
| ≥ 7 days | 18 (43.9) | 84 (53.8) | Ref | | | |

*(Continued)*

**Table 4.** (Continued)

| Characteristics | Aspergillus seropositivity | | Univariate | | Multivariate | |
|---|---|---|---|---|---|---|
| | Yes | No | Crude OR (95% CI) | p-Value | Adjusted OR (95% CI) | p-Value |
| **Fever** | | | | | | |
| Yes | 21 (51.2) | 96 (61.5) | 0.66 (0.33–1.31) | 0.231 | | |
| No | 20 (48.8) | 60 (38.5) | Ref | | | |
| **Cough** | | | | | | |
| Yes | 27 (65.9) | 108 (69.2) | 0.86 (0.41–1.78) | 0.679 | | |
| No | 14 (34.1) | 48 (30.8) | Ref | | | |
| **Weight loss** | | | | | | |
| Yes | 25 (61.0) | 119 (76.3) | 0.49 (0.24–1.01) | **0.049** | 0.34 (0.13–0.88) | **0.027** |
| No | 16 (39.0) | 37 (23.7) | Ref | | Ref | |
| **Dyspnea** | | | | | | |
| Yes | 18 (43.9) | 72 (46.2) | 0.91 (0.46–1.83) | 0.797 | | |
| No | 23 (56.1) | 84 (53.8) | Ref | | | |
| **Pleuritic chest pain** | | | | | | |
| Yes | 8 (19.5) | 38 (24.4) | 0.75 (0.32–1.77) | 0.514 | | |
| No | 33 (80.5) | 118 (75.6) | Ref | | | |
| **Night sweats** | | | | | | |
| Yes | 5 (12.2) | 39 (25.0) | 0.42 (0.15–1.14) | 0.080 | 0.39 (0.12–1.26) | 0.115 |
| No | 36 (87.8) | 117 (75.0) | Ref | | Ref | |
| **Hemoptysis** | | | | | | |
| Yes | 9 (22.0) | 26 (16.7) | 1.41 (0.60–3.29) | 0.431 | | |
| No | 32 (78.0) | 130 (83.3) | Ref | | | |

Most of the CPA patients in this study experienced one or more chronic respiratory symptoms. Common symptoms were weight loss (65.9%), cough (61.0%) and dyspnea (43.5%), but none was associated with *Aspergillus* seropositivity or CPA. Nonetheless, hemoptysis, which is one of the most dramatic and cardinal symptoms of CPA as described by Sapienza was present in only 22% of the patients [28]. The lack of distinctive clinical features clearly indicates that early detection of CPA in patients with chronic respiratory symptoms requires detection of *Aspergillus* antibody and compatible radiologic findings for diagnosis [29,30]. Because fever is inherently an uncommon symptom of CPA, the high proportion of febrile patients with *Aspergillus* seropositivity reported in this study warrants a thorough investigation of concurrent pneumonia (or bronchitis), which is reported in 65.5% of patients in this study or subacute invasive aspergillosis whenever an acute presentation occurs in patients with CPA [31,32].

In our study, 13.8% of *Aspergillus* antibody-positive patients were HIV-infected and 9.8% had dual CPA/HIV infections, which is different from the 7% reported from India [33]. Although these findings may reflect the high burden of HIV in this setting, people living with HIV were less likely to have seropositive *Aspergillus* antibody on multivariate analysis [34,35]. The reduced ability of HIV-infected individuals to produce antibodies or induce inflammatory changes due to impaired immune responses may explain the protective effect of HIV against *Aspergillus* seropositivity [36]. Notably, many patients with a positive *Aspergillus* antibody and CPA had either been diagnosed with TB in the past or were receiving anti-TB treatment at the time of recruitment. TB was an independent predictor of *Aspergillus* seropositivity and CPA in our univariate analysis and remained significant even in multivariable analysis. CPA is known to coexist with other comorbidities, especially TB. More importantly, nearly 50% of patients

**Table 5. Factors associated with CPA.**

| Characteristics | CPA | | Univariate | | Multivariate | |
|---|---|---|---|---|---|---|
| | Yes | No | Crude OR (95% CI) | p-Value | Adjusted OR (95% CI) | p-Value |
| **Gender** | | | | | | |
| Male | 17 (73.9) | 130 (74.7) | 0.96 (0.36–2.59) | 0.934 | | |
| Female | 6 (26.1) | 44 (25.3) | Ref | | | |
| **Age,** *years* | | | | | | |
| < 45 | 12 (52.2) | 79 (45.4) | 1.31 (0.55–3.13) | 0.540 | | |
| ≥45 | 11 (47.8) | 95 (54.6) | Ref | | | |
| **Body mass index**, kg/m$^2$ | | | | | | |
| < 18.5 | 16 (69.6) | 108 (62.1) | 1.40 (0.55–3.57) | 0.484 | | |
| ≥18.5 | 7 (30.4) | 66 (37.9) | | | | |
| **Relationship status** | | | | | | |
| Single | 4 (17.4) | 59 (33.9) | 0.41 (0.13–1.26) | 0.110 | 0.51 (0.16–1.70) | 0.275 |
| Married/others | 19 (82.6) | 115 (66.1) | Ref | | Ref | |
| **Highest education attained** | | | | | | |
| None | 5 (21.7) | 41 (23.6) | 0.90 (0.31–2.58) | 0.846 | | |
| Primary or higher | 18 (78.3) | 133 (76.4) | Ref | | | |
| **Smoking** | | | | | | |
| Yes | 6 (26.1) | 75 (43.1) | 0.47 (0.18–1.24) | 0.119 | 0.38 (0.13–1.17) | 0.092 |
| No | 17 (73.9) | 99 (56.9) | Ref | | Ref | |
| **Alcohol use** | | | | | | |
| Yes | 10 (43.5) | 67 (38.5) | 1.23 (0.51–3.00) | 0.646 | | |
| No | 13 (56.5) | 107 (61.5) | Ref | | | |
| **Drug use** | | | | | | |
| Yes | 1 (4.3) | 7 (4.0) | 1.08 (0.13–9.24) | 0.941 | | |
| No | 22 (95.7) | 167 (96.0) | Ref | | | |
| **HIV status** | | | | | | |
| Positive | 2 (9.1) | 50 (30.9) | 0.22 (0.05–1.00) | **0.042** | 0.24 (0.05–1.14) | 0.073 |
| Negative | 20 (90.9) | 112 (69.1) | Ref | | Ref | |
| **CD4 count**, cells/mm$^3$ | | | | | | |
| < 200 | 3 (75.0) | 43 (86.0) | 0.49 (-.04–5.38) | 0.551 | | |
| ≥200 | 1 (25.0) | 7 (14.0) | Ref | | | |
| **Past or current tuberculosis** | | | | | | |
| Yes | 18 (78.3) | 87 (50.0) | 3.60 (1.28–10.13) | **0.011** | 3.71 (1.23–11.24) | **0.020** |
| No | 5 (21.7) | 87 (50.0) | Ref | | Ref | |
| **Asthma** | | | | | | |
| Yes | 1 (4.3) | 7 (4.0) | 1.08 (0.13–9.24) | 1.000 | | |
| No | 22 (95.7) | 167 (96.0) | Ref | | | |
| **COPD** | | | | | | |
| Yes | - | 8 (4.6) | - | - | | |
| No | 23 (100) | 166 (95.4) | | | | |
| **Presumed bacterial pneumonia or bronchitis** | | | | | | |
| Yes | 15 (65.2) | 114 (65.5) | 0.99 (0.40–2.46) | 0.977 | | |
| No | 8 (34.8) | 60 (34.5) | Ref | | | |
| **Duration of respiratory symptoms,** *days* | | | | | | |
| < 7 days | 13 (56.5) | 82 (47.1) | 1.46 (0.61–3.50) | 0.397 | | |
| ≥ 7 days | 10 (43.5) | 92 (52.9) | Ref | | | |

*(Continued)*

**Table 5.** (Continued)

| Characteristics | CPA | | Univariate | | Multivariate | |
|---|---|---|---|---|---|---|
| | **Yes** | **No** | **Crude OR (95% CI)** | **p-Value** | **Adjusted OR (95% CI)** | **p-Value** |
| **Fever** | | | | | | |
| Yes | 13 (56.5) | 104 (59.8) | 0.88 (0.36–2.11) | 0.766 | | |
| No | 10 (43.5) | 70 (40.2) | Ref | | | |
| **Cough** | | | | | | |
| Yes | 14 (60.9) | 121 (69.5) | 0.68 (0.28–1.67) | 0.400 | | |
| No | 9 (39.1) | 53 (30.5) | Ref | | | |
| **Weight loss** | | | | | | |
| Yes | 12 (52.2) | 132 (75.9) | 0.35 (0.14–0.84) | **0.016** | 0.41 (0.15–1.13) | 0.084 |
| No | 11 (47.8) | 42 (24.1) | Ref | | Ref | |
| **Dyspnea** | | | | | | |
| Yes | 10 (43.5) | 80 (46.0) | 0.90 (0.38–2.17) | 0.821 | | |
| No | 13 (56.5) | 94 (54.0) | Ref | | | |
| **Pleuritic chest pain** | | | | | | |
| Yes | 5 (21.7) | 41 (23.6) | 0.90 (0.32–2.58) | 0.846 | | |
| No | 18 (78.3) | 133 (76.4) | Ref | | | |
| **Night sweats** | | | | | | |
| Yes | 2 (8.7) | 42 (24.1) | 0.30 (0.07–1.33) | 0.095 | 0.39 (0.08–1.90) | 0.241 |
| No | 21 (91.3) | 132 (75.9) | Ref | | Ref | |
| **Hemoptysis** | | | | | | |
| Yes | 4 (17.4) | 31 (17.8) | 0.97 (0.31–3.06) | 0.960 | | |
| No | 19 (82.6) | 143 (82.2) | Ref | | | |

treated for TB in this hospital have Xpert/smear-negative TB [14]. CPA may be the underlying disease in these patients, emphasizing the need to integrate management and prevention of pulmonary fungal infections, including CPA into TB care. Asthma has a well-established association with *Aspergillus* infection [37,38] and in our study, it predicts *Aspergillus* seropositivity in multivariable analysis. However, we did not confirm the presence of fungal asthma in any of the patients due to the limited diagnostic capacity locally [36,37]. Like asthma, COPD has a link with *Aspergillus* infection but was not predictive of seropositivity in our study [39].

The radiographic hallmarks of CPA in patients with chronic lung disease are cavities with or without intracavity fungal ball formation, pleural thickening, and pulmonary fibrosis [40]. We found no patient with a visible fungal ball on chest X-ray, but we didn't have resource to do a computed axial tomography (CT) scan, which would probably have been more sensitive. In the present study, hilar lymphadenopathy and consolidation was less frequent in CPA, whereas extensive pulmonary fibrosis peri-cavitary fibrosis, pleural thickening and pleural effusion were all more likely to be reported in patients with *Aspergillus* seropositivity and thus could provide valuable information in the diagnosis of CPA. Since CT, which provides better lung details, was not used to diagnose CPA, we may have underestimated the true prevalence of CPA in this study.

Our study used a relatively small sample size and was conducted in one center, so the findings cannot be easily generalized to the Sierra Leonean population with chronic respiratory symptoms. Despite these inherent limitations, we provide the first evidence on *A*spergillus seropositivity and CPA in Sierra Leone to highlight the need to train healthcare workers and intensify the diagnosis, management and prevention of CPA in a high TB burden country.

## Conclusion

We report a high prevalence of *Aspergillus* antibody seropositivity and CPA. A common independent predictor of *Aspergillus* seropositivity and CPA was present or past TB infection. These findings underscore the need to integrate the prevention and control of pulmonary fungal infections with TB services and asthma care in order to reduce unnecessary morbidity and mortality.

## Supporting information

**S1 Data. SPSS data set for aspergillus seropositivity and chronic pulmonary aspergillosis.** (SAV)

## Acknowledgments

The data collection was supported by Umu Barrie, Mariama Kamara, Joseph Sandy and Isata Kamara.

## Author Contributions

**Conceptualization:** Sulaiman Lakoh, Joseph B. Kamara, Emma Orefuwa, Darlinda F. Jiba, David W. Denning.

**Data curation:** Daniel Sesay, Darlinda F. Jiba, Maxwell Joseph Kargbo.

**Formal analysis:** George A. Yendewa.

**Methodology:** Sulaiman Lakoh, Joseph B. Kamara, Emma Orefuwa, David W. Denning.

**Resources:** Sulaiman Lakoh, Joseph B. Kamara, Emma Orefuwa, David W. Denning.

**Supervision:** Sulaiman Lakoh, Abubakarr Bailor Bah.

**Writing – original draft:** Sulaiman Lakoh, Emmanuel Firima, George A. Yendewa.

**Writing – review & editing:** Sulaiman Lakoh, Olukemi Adekanmbi, Gibrilla F. Deen, James B. W. Russell, George A. Yendewa, David W. Denning.

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
