## [Decision Letter · Decision Letter 0]

2 May 2023

Dear Dr. Lakoh,

Thank you very much for submitting your manuscript "Prevalence and predictors of Aspergillus seropositivity and chronic pulmonary aspergillosis in an urban tertiary hospital in Sierra Leone: a cross-sectional study" for consideration at PLOS Neglected Tropical Diseases. As with all papers reviewed by the journal, your manuscript was reviewed by members of the editorial board and by several independent reviewers. The reviewers appreciated the attention to an important topic. Based on the reviews, we are likely to accept this manuscript for publication, providing that you modify the manuscript according to the review recommendations. When submitting the revision, please clearly address the comments from reviewer 1 in a point-to-point manner.

Sincerely,

Chaoyang Xue, Ph.D.

Academic Editor

Marcio Rodrigues

Section Editor

Reviewer's Responses to Questions

**Key Review Criteria Required for Acceptance?**

**Methods**

-Are the objectives of the study clearly articulated with a clear testable hypothesis stated?

-Is the study design appropriate to address the stated objectives?

-Is the population clearly described and appropriate for the hypothesis being tested?

-Is the sample size sufficient to ensure adequate power to address the hypothesis being tested?

-Were correct statistical analysis used to support conclusions?

-Are there concerns about ethical or regulatory requirements being met?

Reviewer #1: In this work, Lakoh and colleagues have undertaken a study to assess the predictors of chronic pulmonary aspergillosis in Sierra Leone. The author's have tested Aspergillus seropositivity and CPA diagnosis defined by the GAFFI panel. The authors then used the demographic, clinical and radiographic variables for their utility as CPA predictors. The authors have identified current or prior TB as an independent predictor for CPA for their patient cohort in Sierra Leone. 

Overall, the employed methods are sound, however, this reviewer wondered why other underlying conditions such as ABPA, asthma, pneumonia were also not considered in the authors' analyses. As it has been documented previously that the aforementioned conditions can be predisposing to CPA, can these conditions can also serve as predictors?

Reviewer #2: There are no concerns about the ethical or regulatory requirements being met and statistical analyses are correctly applied. The sample size collected was actually larger than required based on power analysis and the population of patients recruited is clearly described.

**Results**

-Does the analysis presented match the analysis plan?

-Are the results clearly and completely presented?

-Are the figures (Tables, Images) of sufficient quality for clarity?

Reviewer #1: The authors have described the results succinctly, however, the tables are organized in a confusing way; for examples, if footnotes are provided regarding how the data are organized, that would help the readers. Furthermore, Table 3 requires corrections; it is missing values.

Reviewer #2: The analyses exactly match the proposed plan and all results are clearly described. The figure and tables are high quality and arrange the data logically.

**Conclusions**

-Are the conclusions supported by the data presented?

-Are the limitations of analysis clearly described?

-Do the authors discuss how these data can be helpful to advance our understanding of the topic under study?

-Is public health relevance addressed?

Reviewer #1: The authors described current or prior TB as an independent predictor for CPA for their patient cohort, however, these findings are to be expected considering several previous studies that the authors have cited. The authors discuss their findings in light of the previous findings in Kenya, Ghana and Nigeria; however, it would have been helpful if the authors could discuss other chronic respiratory disorders, in addition to TB or non-tuberculous conditions, and their potential roles towards predicting TB.

Reviewer #2: The conclusions are clearly supported by the data presented and limitations and alternative interpretations are provided. The authors provide an excellent Discussion section addressing not only the interpretations of the data but also how it advances our understanding and what aspect will need to be further studied.

**Editorial and Data Presentation Modifications?**

Reviewer #1: (No Response)

Reviewer #2: None

**Summary and General Comments**

Reviewer #1: Overall, this study aims to characterize CPA prevalence and its predictors in Sierra Leone. In the light of increasing awareness towards fungal diseases, this study serves a critical unmet need; however, considering the previously established links between TB and CPA by multiple groups, the lack of novelty decreases this reviewer's enthusiasm.

Reviewer #2: The data were appropriately collected and well-described. The manuscript is well-written with clear interpretations of the data and alternatives are provided. Conclusions are supported by the data. This is a timely study reporting findings from an understudied area.

PLOS authors have the option to publish the peer review history of their article (what does this mean?). If published, this will include your full peer review and any attached files.

Reviewer #1: No

Reviewer #2: No

Figure Files:

Data Requirements:

Please note that, as a condition of publication, PLOS' data policy requires that you make available all data used to draw the conclusions outlined in your manuscript. Data must be deposited in an appropriate repository, included within the body of the manuscript, or uploaded as supporting information. This includes all numerical values that were used to generate graphs, histograms etc.. For an example see here: http://www.plosbiology.org/article/info:doi%2F10.1371%2Fjournal.pbio.1001908#s5.

Reproducibility:

References

---

## [Editor Report · Decision Letter 1]

14 May 2023

Dear Dr. Lakoh,

We are pleased to inform you that your manuscript 'Prevalence and predictors of Aspergillus seropositivity and chronic pulmonary aspergillosis in an urban tertiary hospital in Sierra Leone: a cross-sectional study' has been provisionally accepted for publication in PLOS Neglected Tropical Diseases.

Best regards,

Chaoyang Xue, Ph.D.

Academic Editor

Marcio Rodrigues

Section Editor

---

## [Editor Report · Acceptance letter]

13 Jul 2023

Dear Dr. Lakoh,

We are delighted to inform you that your manuscript, "Prevalence and predictors of *Aspergillus* seropositivity and chronic pulmonary aspergillosis in an urban tertiary hospital in Sierra Leone: a cross-sectional study," has been formally accepted for publication in PLOS Neglected Tropical Diseases.

Best regards,

Shaden Kamhawi

co-Editor-in-Chief

Paul Brindley

co-Editor-in-Chief
